# Emergence of New Immunopathogenic Factors in Human Yellow Fever: Polarisation of the M1/M2 Macrophage Response in the Renal Parenchyma

**DOI:** 10.3390/v14081725

**Published:** 2022-08-04

**Authors:** Juliana Marinho Melo, Luiz Fabio Magno Falcão, Lucas Coutinho Tuma da Ponte, Camilla Costa Silva, Livia Caricio Martins, Jannifer Oliveira Chiang, Arnaldo Jorge Martins Filho, Edna Cristina Santos Franco, Maria Irma Seixas Duarte, Jorge Rodrigues de Sousa, Pedro Fernando da Costa Vasconcelos, Juarez Antônio Simões Quaresma

**Affiliations:** 1Secção de Arbovirologia e Febres Hemorragicas, Instituto Evandro Chagas, Ananindeua 67015-120, PA, Brazil; 2Departamento de Patologia, Universidade do Estado do Pará, Belém 66550-040, PA, Brazil; 3Faculdade de Medicina, Universidade de São Paulo, São Paulo 05508-570, SP, Brazil; 4Núcleo de Medicina Tropical, Universidade Federal do Pará, Belém 66075-110, PA, Brazil

**Keywords:** yellow fever, kidney, M1 macrophage, M2 macrophage, immunopathogenesis

## Abstract

Macrophages in the kidney play a pathogenic role in inflammation and fibrosis. Our study aimed to understand the polarisation of the M1 and M2 phenotypic profiles of macrophages in injured kidney tissue retrieved from fatal cases of yellow fever virus (YFV). A total of 11 renal tissue biopsies obtained from patients who died of yellow fever (YF) were analysed. To detect antibodies that promote the classical and alternative pathways of macrophage activation, immunohistochemical analysis was performed to detect CD163, CD68, inducible nitric oxide synthase (iNOS), arginase 1, interleukin (IL)-4, IL-10, interferon (IFN)-γ, IFN-β, tumour necrosis factor (TNF)-α, IL-13, and transforming growth factor (TGF)-β. There was a difference in the marker expression between fatal cases of YFV and control samples, with increased expression in the cortical region of the renal parenchyma. The immunoexpression of CD68 and CD163 receptors suggests the presence of activated macrophages migrating to infectious foci. The rise in IL-10, IL-4, and IL-13 indicated their potential role in the inactivation of the inflammatory macrophage response and phenotypic modulation of M2 macrophages. The altered expression of IFN-γ and IFN-β demonstrates the importance of the innate immune response in combating microorganisms. Our findings indicate that the polarisation of M1 and M2 macrophages plays a vital role in the renal immune response to YFV.

## 1. Introduction

Flaviviruses are of great epidemiological importance worldwide because they cause diseases in humans and animals of economic interest. Furthermore, they have a wide geographic distribution capable of causing large epidemics. Among them, the YFV is a flavivirus belonging to the Flaviviridae family of the Riboviria kingdom [1,2].

The epidemiological importance of the disease is due to its high dissemination potential, risk of re-urbanisation of transmission, and clinical severity, with a fatality rate of approximately 50% among severe cases and has historically been considered to greatly impact medicine and public health [3].

Macrophages and dendritic cells are crucial early targets of infection. Macrophages can be influenced by a wide variety of tissue factors that alter their phenotype and function. Activated macrophages are generally divided into two categories, M1-type macrophages and M2-type macrophages. The functions of M1 and M2 macrophages are closely related to inflammatory responses, among which M1 macrophages are mainly involved in pro-inflammatory responses and M2 macrophages are mainly involved in anti-inflammatory responses [4]. These cells support productive viral replication and can travel to local lymph nodes and other tissues and organs, promoting systemic spread. Macrophage infection also results in intense cytokine production, commonly referred to as a cytokine storm. This can promote vascular leakage and hypotension and activate coagulation pathways, ultimately leading to disseminated intravascular coagulation. Furthermore, cytokines are likely to contribute to lymphocyte apoptosis. The infection of dendritic cells leads to a dysregulated phenotype, wherein interferon (IFN) responses are suppressed, and the maturation of dendritic cells is impaired. This likely inhibits T-cell activation, further preventing infection control [5].

Renal alterations are secondary to liver damage but culminate in tissue damage that results in an increased level of nitrogenous excreta, such as urea and creatinine, which present in high levels in the plasma circulation. Changes in the kidneys result in a decrease in glomerular filtration capacity, culminating in oliguria or anuria, which worsens retention and may also result in acute renal tubule damage, the exacerbation of retention, and oligoanuria, eventually leading to acute renal failure [6,7]. Thus, the objective of our study was to understand, through immunohistochemistry, the polarisation of the M1 and M2 phenotypic profiles of macrophages in the injured kidney tissue of fatal cases affected by the YFV and its possible relationship with the lesions and/or renal alterations observed during severe yellow fever.

## 2. Methods

In total, 11 renal tissue samples were obtained from patients previously diagnosed with fatal cases of YFV infection by real-time PCR from the archives of the pathology section of the Instituto Evandro Chagas (Belém, Brazil). Additional samples from patients previously diagnosed by RT-PCR were obtained by viscerotomy, fixed in 10% buffered formalin, and embedded in paraffin. The biopsies were sectioned on a microtome at 3 μm thickness.

The patients were between 15 and 63 years old (mean age = 37 years), and 90% were men. The patients came from the states of Tocantins, Goiás, Distrito Federal, and Paraíba, where most cases of infection came from the state of Goiás between 2000 and 2016. Detailed information regarding the sampling included in this study can be found in Table 1.

The control group (n = 3) included kidney specimens from patients who died without any infectious disease or kidney injury, confirmed by histological evaluation, clinical history, and lack of flavivirus positivity, and with no record of other diseases with primary or secondary renal impairment.

### 2.1. Real-Time PCR for Yellow Fever Diagnosis

Biological samples (blood and frozen liver tissue samples) were subjected to diagnostic tests to detect YFV by real-time PCR according to the protocol described by Domingo et al. [8]. The tests used specific primers and probes for the 5′ NCR region (Table 2) of YFV genome common to the seven virus genotypes, with a detection limit of 10.22 copies per reaction. The signal from the fluorescent dye (FAM^®^) at the 5′ end of the YFV-specific probe was inhibited by the TMR at the 3′ end. The analysis of exogenous control was performed based on primers and probes developed by Menting et al. [9], which are specific to the region of the MS2 phage genome that encodes the maturation protein (protein A) as described in Table 2. The fluorescent dye signal (VIC^®^) at the 5′-end of the MS2-specific probe was inhibited by BHQ1 at the 3′-end. The samples were processed using the 7500 Fast Real-Time PCR System (Applied Biosystems, Waltham, MA, USA) and AriaMx Real-Time PCR System (Agilent Technologies, Santa Clara, CA, USA) using two RT-qPCR kits: (1) Superscript III^®^ Platinum^®^ One-Step Quantitative RT-PCR System (Invitrogen, Waltham, MA, USA) and (2) QuantiTect^®^ Probe RT-PCR (Qiagen, Germantown, MD, USA).

### 2.2. Immunohistochemistry Technique for M1 and M2 Macrophage Markers

Paraffin-embedded histological sections were processed for histopathology and stained with hematoxylin and eosin (HE). For immunohistochemistry (IHC), an adapted streptavidin alkaline phosphatase (SAAP) assay with anti-YFV polyclonal mouse antibody. The polyclonal anti-YFV antibody used in the IHC assay was prepared at the Instituto Evandro Chagas (IEC, Ananindeua, Brazil) in young Swiss mice using a YFV strain isolated in cell culture (C6/36 cells).

For the characterisation of M1 and M2 macrophages, paraffin-embedded sections were incubated 14 h at 4 °C with primary monoclonal antibodies diluted in 1% bovine serum albumin (Table 3). After two washes in PBS (pH 7.4) for 3 min each, the samples were incubated with the biotinylated secondary antibody LSAB (BIOTIN-REVEAL DCMT-125/SPRING) in an oven for 30 min at 37 °C. After the first incubation, the slides were washed again in PBS (pH 7.4) for 3 min and incubated with streptavidin peroxidase (REVEAL HRP CONJUGATE DHRR-125/SPRING) for 30 min at 37 °C.

After two washes in PBS (pH 7.4), the reactions were developed using 0.03% diaminobenzidine in 3% hydrogen peroxide as chromogen solution. The specimens were washed in distilled water for 5 min, counterstained with Mayer’s haematoxylin for 2 min, washed again in distilled water, dehydrated in ethanol, cleared in xylene, and mounted in Permount diluted with coverslips for further reading and analysis.

### 2.3. Quantitative Analysis

A Zeiss Axio Imager Z1 microscope (template: 456006, Oberkochen, Germany) was used for the analysis. Immunomarkers were quantitatively analysed by selecting 10,400× fields of the renal parenchyma. Each field was subdivided into 10 × 10 areas delimited by the microscope lens, comprising a region of 0.0625 mm^2^.

### 2.4. Statistical Analysis

The results obtained were stored in electronic spreadsheets using Excel 2019 (Microsoft Corporation, Redmond, WA, USA) and analysed using GraphPad Prism 8.0^TM^ (Graphpad software, Inc., San Diego, CA, USA). The hypotheses were tested using non-parametric tests, the Mann–Whitney test, and Spearman’s rank correlation coefficient. Data are presented as mean ± standard deviation (SD). Significance was set at *p* ≤ 0.05.

## 3. Results

The histopathology of the kidney showed congestion, interstitial edema, glomerular congestion and necrosis, swelling of the tubular lining epithelium and acute tubular necrosis. IHC for the YFV showed the presence of viral antigen mainly in the tubular epithelium (Figure 1), which shows the presence of the virus antigen in the renal parenchyma accompanied by mild to moderate nephritis.

Quantitative analysis of the phenotypic markers of M1/M2 macrophages in the renal parenchyma between the samples of fatal cases of human YFV and control was performed using markers with monoclonal antibodies against CD163, CD68, inducible nitric oxide synthase (iNOS), arginase 1 (Arg1), interleukin (IL)-4, IL-10, IFN-γ, IFN-β, tumour necrosis factor-alpha (TNF-α), IL-13, and transforming growth factor (TGF-β). There was a statistically significant increase in the expression of CD68, CD163, iNOS, Arg1 enzymes, and cytokines in the renal parenchyma of case samples compared to controls (*p* ≥ 0.05), as shown in Figure 2.

Among the analysed cases, the average number of CD68-labelled cells was 28 ± 17.26 cells per field, with an increase in the statistically significant difference compared to control cases (*p* = 0.0082) (Figure 2A). The mean number of CD163-labelled cells was 39.45 ± 24.60 cells per field, which was significantly different from that in control cases (*p* = 0.0055) (Figure 2B). For iNOS expression, we observed 71.91 ± 25.02 cells per field, with a statistically significant difference compared to control cases (*p* = 0.0055) (Figure 2C). In contrast, the mean number of Arg1-labelled cells per field was 38.82 ± 22.65, which was significantly different from that in control cases (*p* = 0.0082) (Figure 2D).

Regarding anti-inflammatory cytokines, immunostaining revealed that the mean number of IL-4-labelled cells per field was 31.55 ± 8.96, which was significantly different from that in control cases (*p* = 0.0055) (Figure 2E). The mean number of cells labelled for IL-10 was 40.36 ± 31.25 cells per field, with a significant difference compared to that in the control cases (*p* = 0.0055) (Figure 2F). In contrast, the mean number of IL-13-labelled cells was 35.91 ± 11.84 cells per field, with a statistically significant difference compared to that in control cases (*p* = 0.0027) (Figure 2G).

The immunostaining results for IFN-γ revealed that the average number of labelled cells was 37.18 ± SD 20.85 cells per field, with a statistically significant difference compared to the control cases (*p* = 0.0082) (Figure 2H). In the immunostaining for IFN-β, the average number of labelled cells was 28.73 ± SD 7.78 cells per field, with a statistically significant difference compared to control cases (*p* = 0.0055) (Figure 2I).

The mean number of cells labelled for TGF-β was 28.73 ± SD 7.78 cells per field, with a statistically significant difference compared to control cases (*p* = 0.0055) (Figure 2J). Immunostaining for TNF-α revealed that the average number of cells labelled was 52.09 ± SD 20.85 cells per field, with a statistically significant difference compared to control cases (*p* = 0.0055) (Figure 2K). The graphs obtained from this analysis, composed of the mean of the analysed immunomarkers, are shown in Figure 2A–K.

The immunostaining pattern demonstrated that positive areas in the tissue were characterized by the deposition of a brown precipitate of nuclear, cytoplasmic, or extracellular distribution according to each antigen investigated, as seen in Figure 3A–K.

## 4. Discussion

Yellow fever is endemic in tropical regions of South America and Sub-Saharan Africa and is categorised as a re-emerging disease due to its great risk to public health. The clinical presentation includes a classic viral haemorrhagic fever with a high fatality rate, clinically manifested as liver dysfunction, acute renal failure, coagulopathy, and shock [2,10]. During the clinical course of acute, bleeding symptoms and fatal outcomes are strongly correlated with highly elevated pro- and anti-inflammatory cytokines, suggesting an immune contribution to the disease pathogenesis [11].

Although the detailed role of cytokines has not been fully elucidated, hepatocytes [12], endothelial cells [13], and activated macrophages [14] may contribute to the clinical course of the disease. Furthermore, immunological elimination of YFV can exacerbate its pathogenesis [15]. Some studies have demonstrated the presence of the virus in the urine and semen even after the negative results of molecular biology tests for the virus in the blood, which demonstrates that the kidney is one of the target organs compromised by the virus, despite the few studies in the literature [16,17]. Proteinuria also promotes the recruitment of macrophages, which perpetuate the interstitial inflammatory reaction, resulting in interstitial nephritis [18].

Cytokines may also be elevated, such as IL-6, IFN-γ-induced IL-10, monocyte chemoattractant protein 1 (MCP-1), and TNF-α, which constitute immune mediators of inflammatory reactions and are not necessarily specific for YFV [19].

The polarisation of pro-inflammatory macrophages (M1) occurs by various immune mediators, such as IFN-γ, which are released by neighbouring inflammatory cells, including neutrophils, natural killer cells, and effector T cells (predominantly Th1/Th17). Activated M1 macrophages can further exacerbate tissue inflammation and cause kidney damage [20].

Subsequently, Th2 lymphocytes and regulatory T cells are recruited to the kidney and regulate immune responses. This includes shifting macrophages to an anti-inflammatory (M2) phenotype and releasing regulatory cytokines, such as IL-13, IL-10, and IL-4. M2 macrophages predominate at this stage and contribute to the resolution of inflammation and tissue repair. Fibrosis can occur depending on the severity of the lesion and whether pathogenic factors are expressed [21].

Quantitative analysis of the markers used in our study showed a significant increase in these cytokines, mainly TNF-α, followed by IFN-γ, which participates in the viral clearance process, potentiating inflammation and the consequent manifestation of the Th1 response. This is reflected in the activation of macrophages and other cells, increasing the expression of MHC class I and II molecules and the response of inhibition of viral replication in the renal parenchyma.

Associated with the aforementioned findings, statistically significant immunoexpression of the anti-inflammatory cytokines, IL-13, IL-10, and IL-4, suggests the contribution of a regulatory response that would reduce inflammation and repair of renal tissues.

The immunoexpression of TGF-β, a cytokine with immunosuppressive potential, was similar to its activity in the liver and is possibly associated with the cellular immune response characteristic of this infection [11]. The entire process of kidney injury, including proteinuria and the reabsorption of protein by the proximal tubules, causes these cells to release cytokines and chemokines that act by recruiting immune cells to the site of renal inflammation.

Tubular cells react to this inflammatory process through the glomerular and tubular epithelial–mesenchymal transition mediated by cytokines and transform into interstitial fibroblasts. In addition, they promote the derangement of the basement membrane by proteases originating from the affected epithelial cells of the glomerulus. Proteins from the TGF-β superfamily, particularly TGF-β and bone morphogenetic protein (BMP-7), are the main mediators of this process [21,22,23].

TGF-β stimulates the epithelial–mesenchymal transition and the synthesis of extracellular matrix proteins such as collagens and fibronectin. BMP-7 inhibits epithelial–mesenchymal transition and stimulates the synthesis of proteolytic enzymes, such as matrix metalloproteinases 2 and 9 [21,22,23].

We believe that the iNOS and Arg1 response observed by immunostaining influences the response pattern of M1 and M2 macrophages, as both enzymes compete for L-arginine to induce the production of nitric oxide (NO) and growth factors in the investigated fatal cases [24].

Arg1 converts L-arginine into urea and L-ornithine in the final step in the urea cycle. The resulting polyamines are crucial for cell proliferation and removing toxins resulting from protein degradation. Arg1 deprives NO of its substrate synthesis by degrading arginine and deregulates NO production. In mice, Arg1 expression is one of the hallmarks of alternatively activated macrophages (M2a). In addition, it is a key effector and marker of M2a macrophages and myeloid cells, which are the main mediators of T-cell suppression [25].

Arg1 is critical in the study of renal tissue inflammation. It is present in humans and mice and can be added to any panel to identify differentiated macrophages and their activation state. In our study, the markers CD68 and CD163 showed significant immunoexpression. Our research demonstrated a significant expression of these markers, CD68 and CD163, indicating the presence of macrophages in the inflammatory infiltrate of injured kidney tissue of patients with YFV.

Notably, we observed differences in positive immunostaining of IFN-β in the control cases compared to that in the fatal cases investigated. IFN-β is a type I interferon produced by fibroblasts. This interferon type has antiviral, antiproliferative, and immunomodulatory effects [26].

Two forms of IFN-β, IFN-β 1a and IFN-β 1b, are used therapeutically, and two different formulations of IFN-1a are available. For example, in multiple sclerosis (MS), IFN-β initiates several complex events that contribute to altered gene transcription, thereby affecting many genetic pathways. Although other pathways may contribute to the therapeutic effects of IFN-β in MS, its effect on immune function is believed to be the most plausible. Within the immune system, IFN-β can reduce antigen presentation, inhibit T-cell proliferation, and alter cytokine production. It can also restore suppressor function, which appears to be impaired in patients with MS [27].

Our findings suggest that this discrepancy referring to the statistical increase in the mean of controls compared to positive cases may have been because these control cases had been administered an alternative form of IFN-β-based therapy. In contrast, immunostaining results reflect the antiviral function of this IFN type as well as the participation of IFN-α in cell control and replication and are modifiers of the immune response. Cells infected with YFV produce and secrete interferons, which bind to specific receptors on neighbouring cells [28].

TNF-α plays a critical role in mediating kidney damage, which has an autocrine function in the activation of macrophages, induces apoptosis, and coordinates the activation of a network of cytokines and chemokines in the kidney, corroborating our findings regarding the quantification of this important factor in the defence mechanism signalling in the body.

Using the immunohistochemistry technique, a study with the Dengue flavivirus showed that the cytokines IL-4, IL-10, IL-13, TNF-α, and IFN-γ were rarely observed in human kidney tissue. However, an excessive number of CD68 macrophages was observed, suggesting that viruses of the same family and genus may elicit varying tissue immune responses [29].

Targeted recruitment of circulating monocytes to the interstitial compartments is critical. These differentiate into M1 or M2 macrophages in renal tissues depending on the local microenvironment signals during tissue infection or injury.

Pro-inflammatory M1 macrophages release inflammatory mediators, including TNF-α and IFN-γ, which stimulate iNOS production and consequently induce tissue inflammation to recognise and eliminate YFV in this tissue environment. In contrast, M2 macrophages release anti-inflammatory mediators, including IL-10, IL-13, and IL-4, which induce the production of Arg1 and TGF-β to repair the tissue injury.

Our collective novel observations from the human kidney tissue samples suggest that the polarisation of M1 and M2 macrophages actively contributed to the renal tissue response to YFV.

## 5. Conclusions

In conclusion, renal tissue injuries may be related to the induction of an M2 macrophage-mediated inflammatory environment, indicating that viral agents, such as YFV, may still benefit from the induction of specific tissue environments related to the Th2 response and, consequently, tissue repair and induction of tissue fibrosis (Figure 4). This factor can be corroborated by identifying YFV in urine samples, even in later stages of infection [30]. However, further studies are warranted to understand the immunopathogenesis of YFV in human kidney tissues.

## Figures and Tables

**Figure 1 viruses-14-01725-f001:**
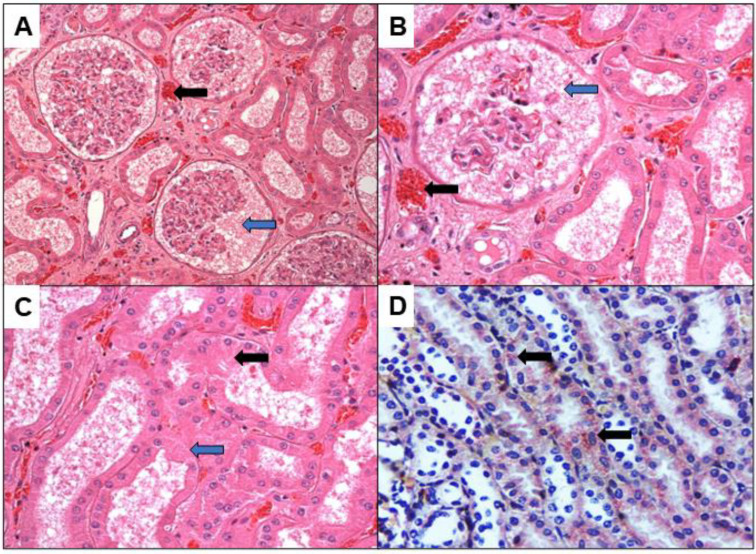
(**A**–**D**) Histopathology of sections stained by HE (**A**–**C**) and IHC method for YFV (**D**) in the kidney of fatal cases of YF. The figure demonstrates renal parenchymal congestion (black arrows in **A**,**B**), necrosis of glomerulus and tubular epithelium ((blue arrows in **A**–**C**). YFV antigen demonstrated in the epithelium of renal tubular structures (black arrows in **D**). Magnification 200× (**A**,**D**) and 400× (**B**,**C**).

**Figure 2 viruses-14-01725-f002:**
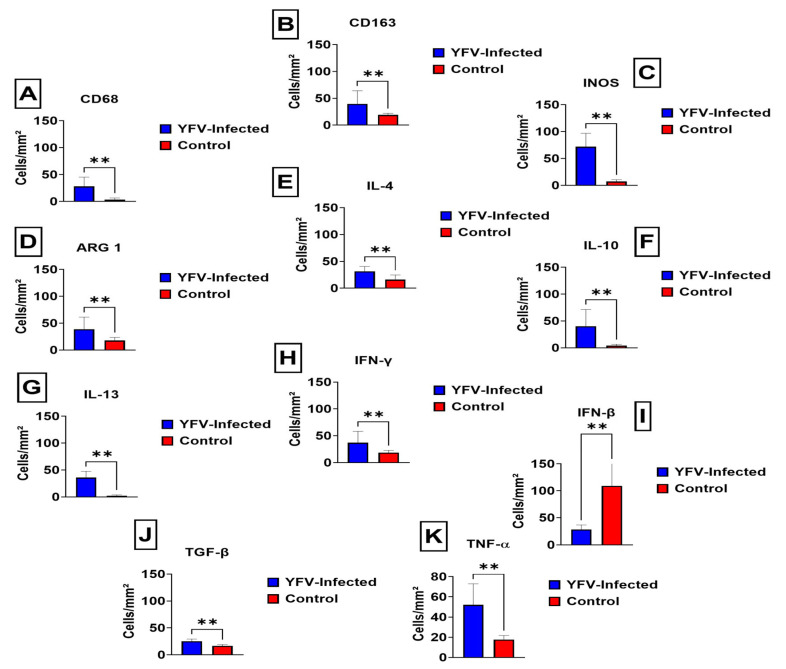
(**A**–**K**) Quantitative analysis of M1 and M2 macrophage polarisation drivers in the renal parenchyma of fatal cases affected by YFV. ** *p* < 0.05.

**Figure 3 viruses-14-01725-f003:**
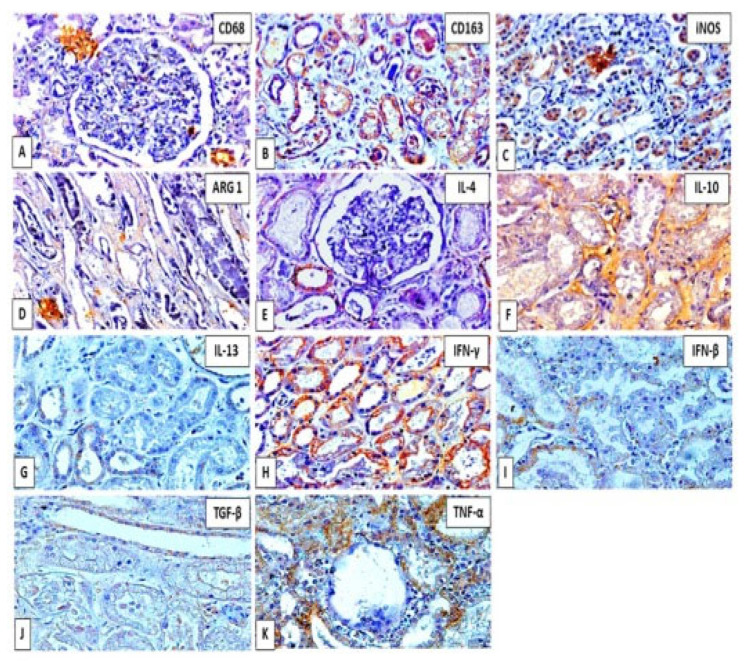
(**A**–**K**): Positive immunostaining pattern of factors that characterize the phenotype of M1 and M2 macrophages in the renal parenchyma of fatal yellow fever cases. Positive areas are characterized by the deposit of brownish material in the nucleus or cytoplasm of cells in glomeruli and tubules. Magnification 200×.

**Figure 4 viruses-14-01725-f004:**
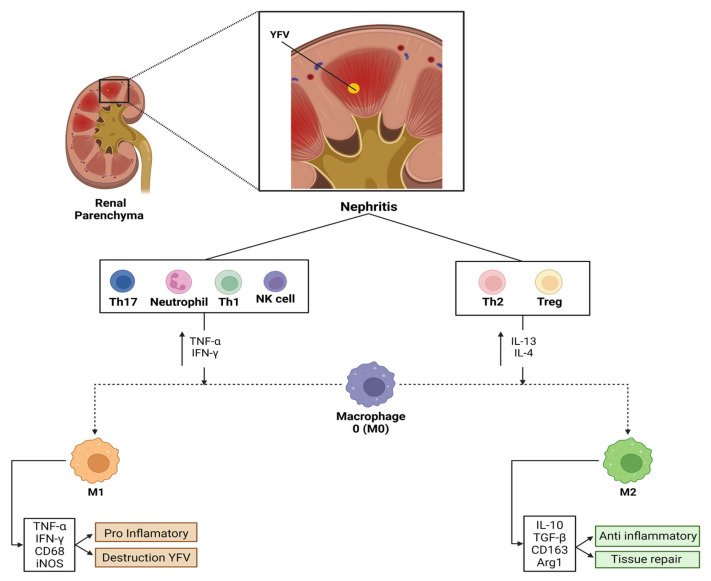
Schematic demonstration of the pathophysiology of renal impairment in YF and its correlation with the role of M1 and M2 macrophages. Note the relationship of the M1 response associated with tissue destruction and M2 macrophages to tissue repair.

**Table 1 viruses-14-01725-t001:** Characterisation of yellow fever patients according to their precedence, age and gender, illness time (I.T.).

Case	State	Sex	Age	I.T. (Days)	Patient	Year
1	TO	M	30	N.I.	494/00	2000
2	GO	M	48	-	255/00	2000
3	GO	M	23	N.I.	074/07	2007
4	GO	F	63	2	043/08	2008
5	DF	M	55	-	088/08	2008
6	GO	M	42	N.I.	095/08	2008
7	DF	M	35	N.I.	154/08	2008
8	GO	M	35	N.I.	062/16	2016
9	PB	M	-	N.I.	102/16	2016
10	GO	M	15	7	346/16	2016
11	GO	M	27	1	369/16	2016

N.I.—not included. (-)—not found, M—Male; F—Female; GO—Goiás; PB—Paraíba; TO—Tocantins; DF—Distrito Federal.

**Table 2 viruses-14-01725-t002:** Primers and probe used in the RT-qPCR technique for the detection of YFV and phage MS2.

**Primers or Probe**	**Sequence (5′-3′)**	**Position**
YFallF	5′-GCTAATTGAGGTGYATTGGTCTGC-3′	15–38
YFallR	5′-CTGCTAATCGCTCAAMGAACG-3′	83–103
YFallP	5′-FAM-ATCGAGTTGCTAGGCAATAAACAC-TMR-3′	41–64
**Primers or Probe**	**Sequence (5′-3′)**	**Position**
MS2-F	5′-ATCAAGTTAGATGGCCGTCTGT-3′	841–862
MS2-R	5′-TAGAGACGACAACCATGCCAAAC-3′	963–941
MS2 probe	5′-VIC-TCCAGACAACGTGCAACATATCGCGACGTATCGTGATATGG-BHQ1-3′	881–921

Source: adapted from Domingo et al., 2012 [7] and Menting et al., 2011 [8].

**Table 3 viruses-14-01725-t003:** Specific monoclonal antibodies used in the samples.

Antibody	Mark/Code	Animal	Batch	Work Dilution
TNF-α	Abcam 6671	rabbit	GR235155-32	1:100
IL-4	Abcam9622	rabbit	GR3174920-9	1:100
IL-13	Abcam 9576	rabbit	GR10654-33	1:100
IL-10	Abcam 34843	rabbit	GR200618-33	1:100
IFN-γ	biorbyt/orb 10878	rabbit	676	1:100
INOS	NOVUS QG18859	rabbit	QG218859	1:100
CD 163	NBP2-36494	mouse	A-2	1:100
ARGINASE I	NBP1-87455	rabbit	A63844	1:100
CD 68	MO814	mouse	00012544	1:100
IFN-β	Abcam140211	rabbit	GR3208814-1	1:100
TGF-beta	Abcam 190503	mouse	GR3183728-7	1:100

## Data Availability

The data used to support the fidings of this study are included within the article.

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
