# Peer review of "Emergence of New Immunopathogenic Factors in Human Yellow Fever: Polarisation of the M1/M2 Macrophage Response in the Renal Parenchyma"

_viruses, 2022, doi:10.3390/v14081725_

Round 1

Reviewer 1 Report

This is an interesting manuscript that investigates immunopathologic factors in the kidney associated with fatal yellow fever (YF) virus infection. The data are based on 11 renal tissue biopsies from YF patients compared to three controls (Line 72). There is no doubt that some on the changes observed are very significant, such as CD68, INOS and IL-13 but others, such as TGF-beta, look less convincing.  In my opinion, the authors should have included more controls to give more statistical power to the results.

The major gap in the paper is that there is no immunohistochemistry (IHC) using an anti- YF virus nonstructural (NS) protein to show how much virus infection there was in the kidneys of the 11 patients. This is potentially very important information to compare to the cellular markers investigated. This should be an easy analysis assuming the authors still have the tissue sections.

Line 72: Please provide more information in the three controls, e.g. age, sex, underlying conditions, reason for death. Were the 3 samples sex and age matched to the YF samples?

Line 74: I assume ‘flavivirus positivity” means “lack of….”

Line 80: Were sera or kidney used for RT-PCR? If it is kidney, then Table 1 suggests only two samples were YF positive; hence the need for IHC using anti-YF NS proteins.

Line 129: Is Table 2 correct as this is the Table showing YF primers.

Lines 172-181: p values are not great. Are these data statistically significant?

Lines 226-231: A comparison of innate response markers in kidney vs liver for YF infection would be very helpful to readers who are not familiar with YF work.

Line 263: “therapeutically”. I assume this does not mean YF. Please clarify as it suggests IFN is used as a therapeutic for YF treatment.

Line 272-273: How can you say this when there are only 3 controls.

Author Response

Response to Reviewer 1

This is an interesting manuscript that investigates immunopathologic factors in the kidney associated with fatal yellow fever (YF) virus infection. The data are based on 11 renal tissue biopsies from YF patients compared to three controls (Line 72). There is no doubt that some on the changes observed are very significant, such as CD68, INOS and IL-13 but others, such as TGF-beta, look less convincing.  In my opinion, the authors should have included more controls to give more statistical power to the results.

The major gap in the paper is that there is no immunohistochemistry (IHC) using an anti- YF virus nonstructural (NS) protein to show how much virus infection there was in the kidneys of the 11 patients. This is potentially very important information to compare to the cellular markers investigated. This should be an easy analysis assuming the authors still have the tissue sections.

Response= We agree. We have included a figure representing the histopathological changes and immunohistochemistry for the YFV in the paper. In addition, some studies such as the one by Barbosa et al. (2018) and Reusken et al. (2017) (PMID: 29058663, PMID: 28855304) have already described the urinary tract as one of the main ways of identifying the virus by PCR, and the investigation of the virus in the urine remains positive even after the detection of the virus in the blood is negative by the technique of molecular biology. In this way, we have no doubt that the kidney is one of the organs certainly compromised by the virus. References to this information were included in the manuscript.

Line 72: Please provide more information in the three controls, e.g. age, sex, underlying conditions, reason for death. Were the 3 samples sex and age matched to the YF samples?

Response= We agree. Controls corresponded to 25% of the total sample, as published in other studies on the disease (PMID: 29311619, PMID: 30121258, PMID: 35704977, PMID: 16872652, PMID: 16278000, PMID: 16219382, PMID: 35746675, PMID: 35056050, PMID: 35805137). In addition, we used kidney samples sent to the Instituto Evandro Chagas of the Ministry of Health of Brazil as a positive for the diagnosis of yellow fever and not all cases include kidney samples. Additionally, the kidney samples used in the control needed to have normal histology and not show positivity for arboviruses or other viruses such as the hepatitis virus, and other viruses of importance in public health in Brazil. General demographic information of controls was included in the manuscript.

Line 74: I assume ‘flavivirus positivity” means “lack of….”

Response= We agree. The text has been corrected as suggested.

Line 80: Were sera or kidney used for RT-PCR? If it is kidney, then Table 1 suggests only two samples were YF positive; hence the need for IHC using anti-YF NS proteins.

Response=The diagnosis of yellow fever cases was made by molecular biology for the virus in the serum and/or frozen liver samples according to the protocol of the Ministry of Health of Brazil. We have changed the table 1 to make the manuscript clearer since we did not perform the RT-PCR technique on all kidney samples, but in blood and liver samples for diagnosis.

Line 129: Is Table 2 correct as this is the Table showing YF primers.

Response= We agree. The text has been corrected as suggested.

Lines 172-181: p values are not great. Are these data statistically significant?

Response= We agree. Although there was significance between the values expressed between the normal and YF control groups, there was no statistical significance in the correlation analysis. We removed the figure and adapted the manuscript to exclude this analysis as requested by reviewer 3.

Lines 226-231: A comparison of innate response markers in kidney vs liver for YF infection would be very helpful to readers who are not familiar with YF work.

Response= We agree. However, in the case of the macrophage lineage, these cells have a characteristic function according to the organ in which they are found, being part of the mononuclear phagocytic system. Thus, comparisons between two tissue microenvironments of different organs do not show a correlation of function, since the compartmentalized immune response is affected by several factors of the tissue itself, such as the extracellular matrix and the function of the organ's own parenchymal cells (please see references PMID: 24048120, PMID: 25861979, PMID: 27171409, PMID: 19054576, PMID: 16719987, PMID: 26898110, PMID: 29844586, PMID: 24003919, PMID: 29664562).

Line 263: “therapeutically”. I assume this does not mean YF. Please clarify as it suggests IFN is used as a therapeutic for YF treatment.

Response=In the context in which IFN has been discussed, it has antiviral action and is important in the immune response against the yellow fever virus. 

Line 272-273: How can you say this when there are only 3 controls.

Response= Several works published in the study of the immunopathogenesis of yellow fever, the relationship between cases and controls is in general of 25% of the samples. This relationship has already been validated in several works developed by our group and we believe it to be adequate. Some of these works are cited in our references. In addition, we used kidney samples sent to the Instituto Evandro Chagas of the Ministry of Health of Brazil as a positive control for the diagnosis of yellow fever and not all cases include kidney samples. Additionally, the kidney samples used in the control needed to have normal histology and not show positivity for arboviruses or other viruses such as the hepatitis virus, and other viruses of importance in public health in Brazil. Thus, the samples suitable for control became scarcer. However, the literature of studies for other arboviruses uses this proportion, which is already validated in the literature (please see PMID: 29311619, PMID: 30121258, PMID: 35704977, PMID: 16872652, PMID: 16278000, PMID: 16219382, PMID: 35746675, PMID: 35056050, PMID: 35805137).

Reviewer 2 Report

This is an interesting, well-written manuscript describing the differences in M!/M2 macrophage populations (as determined by immunohistochemistry) in kidney biopsies obtained from control or YFV-infected individuals.

My primary concerns are the lack of information on what constitutes M1 and M2 macrophages in the introduction. At the minimum, the paragraph L296-300 from the discussion should be moved to the introduction. Additional information would be useful.

Additionally, information is lacking in the Materials and Methods section. 

Section 2.1 RT-PCR How was this performed? In situ on the tissues on the slides? Or were the tissues removed and nucleic acids obtained and run in a thermal cycler? Please describe in more detail.

Section 2.2 should be named immunohistochemistry and should start with L97 unless the PCR and immunos were run on the same samples. If that is clarified, the name of the subsection should represent both. If these are separate subsections, the numbering of the subsequent subsections will need to be changed.

The results section describing correlations and significance need to be substantially rewritten. None of the correlations are significant based upon the listed value of p<0.05. The lowest p value obtained was 0.1005. Additionally, it is stated in L175 and the discussion that there is a significant negative correlation between CD163 and Arg1 which is not the case. r=-0.1011 is essentially a zero correlation and p=0.7652 is far from significant.

The correct abbreviation of "interferon" is "IFN" please correct throughout

L30 Please change "the virus" to "YFV" 

L26 "Increase the IL-10..." please correct this sentence

L44 Please change to "...productive viral replication..."

L60 Delete "yellow fever virus" the abbreviation has already been described

L81 What t ype of biological samples?

L288 Change "most" to "any"

L41 Please change "have" to "has"

Author Response

Response to Reviewer 2

This is an interesting, well-written manuscript describing the differences in M1/M2 macrophage populations (as determined by immunohistochemistry) in kidney biopsies obtained from control or YFV-infected individuals.

My primary concerns are the lack of information on what constitutes M1 and M2 macrophages in the introduction. At the minimum, the paragraph L296-300 from the discussion should be moved to the introduction. Additional information would be useful.

Response= We agree. The text has been corrected as suggested.

Additionally, information is lacking in the Materials and Methods section. 

Section 2.1 RT-PCR How was this performed? In situ on the tissues on the slides? Or were the tissues removed and nucleic acids obtained and run in a thermal cycler? Please describe in more detail.

Response= We agree. The diagnosis of yellow fever cases was made by molecular biology for the virus in the serum and/or frozen liver samples according to the protocol of the Ministry of Health of Brazil. We have changed the table 1 to make the manuscript clearer since we did not perform the RT-PCR technique on all kidney samples, but in blood and liver samples for diagnosis. The text has been corrected to clarify the methodology.

Section 2.2 should be named immunohistochemistry and should start with L97 unless the PCR and immunos were run on the same samples. If that is clarified, the name of the subsection should represent both. If these are separate subsections, the numbering of the subsequent subsections will need to be changed.

Response= We agree. The text has been corrected as suggested.

The results section describing correlations and significance need to be substantially rewritten. None of the correlations are significant based upon the listed value of p<0.05. The lowest p value obtained was 0.1005. Additionally, it is stated in L175 and the discussion that there is a significant negative correlation between CD163 and Arg1 which is not the case. r=-0.1011 is essentially a zero correlation and p=0.7652 is far from significant.

Response= We agree. Although there was significance between the values expressed between the normal and YF control groups, there was no statistical significance in the correlation analysis. We removed the figure and adapted the manuscript to exclude this analysis as requested by reviewer 3.

The correct abbreviation of "interferon" is "IFN" please correct throughout

Response= We agree. The text has been corrected as suggested.

L30 Please change "the virus" to "YFV"

Response= We agree. The text has been corrected as suggested.

L26 "Increase the IL-10..." please correct this sentence

Response= We agree. The text has been corrected as suggested.

L44 Please change to "...productive viral replication..."

Response= We agree. The text has been corrected as suggested.

L60 Delete "yellow fever virus" the abbreviation has already been described (ADJUSTED)

Response= We agree. The text has been corrected as suggested.

L81 What type of biological samples?

Response= The diagnosis of yellow fever is made by RT-PCR on blood samples and frozen liver tissue.

L288 Change "most" to "any"

Response= We agree. The text has been corrected as suggested.

L41 Please change "have" to "has"

Response= We agree. The text has been corrected as suggested.

Reviewer 3 Report

This concise study is to study the polarization of the M1 and M2 phenotypic profiles of macrophages in the injured kidney tissues of fatal cases from yellow fever virus (YFV) infected patients using immunohistochemistry, and its possible relationship with the lesions and/or renal alterations observed during severe yellow fever.

Major comments

(1)   This study only relied on immunohistochemistry of renal biopsies, the inclusion of serum/plasma analysis of the studied cytokines such as IL-4, IL-10, IFN-gamma/beta, TNF-alpha etc. of the subjects may be helpful.

(2)   As mentioned in the manuscript]t, inflammatory cytokine IL-6 and Th17 cytokine IL-17 are important in viral renal inflammation, the immunohistochemical analysis of IL-6 and IL-17 should be included in the study.

(3)   The scale bars should be included in Figure 2.

(4)   As there are no statistical significance (all p > 0.05) in Figure 3, there is not many values for presenting Figure 3 in the manuscript.

(5)   It needs further discussion for the underlying mechanism for the polarization of M0 into both M1 and M2 macrophages in renal impairment of fatal yellow fever diseases.  It is also unclear that whether the polarization of M1 and M2 polarization pattern is different at different stages during the progression of renal impairment (e.g. initial, middle and end stages) in yellow fever infection.

(6)   Apart from M1 and M2, other mentioned related immune effector cells such as T cells, dendritic cells, neutrophils etc. should also be analyzed in renal biopsies for more comprehensive evaluation of the underlying inflammatory/cellular mechanisms of renal damage in fatal yellow fever infection.  

Author Response

Response to Reviewer 3

This concise study is to study the polarization of the M1 and M2 phenotypic profiles of macrophages in the injured kidney tissues of fatal cases from yellow fever virus (YFV) infected patients using immunohistochemistry, and its possible relationship with the lesions and/or renal alterations observed during severe yellow fever.

Major comments

  • This study only relied on immunohistochemistry of renal biopsies, the inclusion of serum/plasma analysis of the studied cytokines such as IL-4, IL-10, IFN-gamma/beta, TNF-alpha etc. of the subjects may be helpful.

Response= Our primary objective is to study the compartmentalized immune response pattern and its relationship with renal pathogenesis in YF. Several studies have shown that the study of the compartmentalized tissue response pattern has a closer relationship with the pathogenesis of infections, including YF, and thus are more suitable for studying this relationship in the tissue where these changes occur (Please see: PMID: 26965109, PMID: 26898110, PMID: 31481953, PMID: 29664562, PMID: 12777598, PMID: 6357500, PMID: 21133813, PMID: 29328785, PMID: 30817236, PMID: 15510086, PMID: 35805137, PMID: 16278000, PMID: 34168651, PMID: 30568659).

  • As mentioned in the manuscript]t, inflammatory cytokine IL-6 and Th17 cytokine IL-17 are important in viral renal inflammation, the immunohistochemical analysis of IL-6 and IL-17 should be included in the study.

Response= We agree, but the main objective of the markers used in the present work is to characterize the response of M1 and M2 macrophages. The chosen panel was aimed at answering the possible role of these macrophage subpopulations in the renal pathogenesis of yellow fever virus infection. The role of cytokinetic profiles such as Th1, Th2, Th17 and Th22 will be discussed in a new study to be published by our group in the future.

  • The scale bars should be included in Figure 2.

Response= We agree. We have included the magnification in the figure legend.

(4)   As there are no statistical significance (all p > 0.05) in Figure 3, there is not many values for presenting Figure 3 in the manuscript.

Response= We agree. Although there was significance between the values expressed between the normal and YF control groups, there was no statistical significance in the correlation analysis. We removed the figure and adapted the manuscript to exclude this analysis.

(5)   It needs further discussion for the underlying mechanism for the polarization of M0 into both M1 and M2 macrophages in renal impairment of fatal yellow fever diseases.  It is also unclear that whether the polarization of M1 and M2 polarization pattern is different at different stages during the progression of renal impairment (e.g. initial, middle and end stages) in yellow fever infection.

Response= We agree. We have included a paragraph in introduction showing the possible role of these two subpopulations in the pathogenesis of inflammation. The evaluation by clinical stage (initial to lethal outcome) of the disease is not possible because biopsy in patients with YF is contraindicated due to coagulation disorders. Finally, our series consists of fatal cases of the disease. Information about the stage of clinical evolution was included in the manuscript.

(6)   Apart from M1 and M2, other mentioned related immune effector cells such as T cells, dendritic cells, neutrophils etc. should also be analyzed in renal biopsies for more comprehensive evaluation of the underlying inflammatory/cellular mechanisms of renal damage in fatal yellow fever infection.  

Response= The subpopulation of macrophages M1 and M2 are subpopulations related to the Th1 and Th2 profiles of immune response. Our objective was to study the possible role of these cells in the renal parenchyma in fatal YF. Macrophages in the kidney are antigen-presenting cells, such as the Kupffer cell in the liver (PMID: 32234529, PMID: 29319160, PMID: 30692665, PMID: 30827512, PMID: 34970275), and are one of the important cells in the tissue immune response against the virus. The markers we used allow us to make these associations with other subpopulations of cells such as Th1 and Th2 lymphocytes and antigen-presenting cells.

Round 2

Reviewer 1 Report

This revision is much improved although I still think the number of controls could be improved.

Reviewer 3 Report

The manuscript has been revised and improved accordingly.